# Treatment of Medication-Related Osteonecrosis of the Jaw (MRONJ) with Er:YaG Laser and Ozone Therapy: A Case Series

Gianluca Porcaro, Paolo Caccianiga *, Ayt Alla Bader and Gianluigi Caccianiga

School of Medicine and Surgery, University of Milano-Bicocca, 20900 Monza, Italy
* Correspondence: p.caccianiga@campus.unimib.it

**Abstract:** The purpose of this study is to evaluate the efficacy of the combination of ozone gel and Er:YAG laser treatment in respect of medication-related osteonecrosis of the jaw (MRONJ) for normal procedures. Consequently, the following techniques are compared in the study: medical therapy (MT); MT + conservative surgery with rotary/piezoelectric instruments; MT + ozone therapy; MT + surgical treatment + laser Er:YAG; and MT + ozone therapy + surgical treatment + laser Er:YAG. Fifty-seven patients with MRONJ stages I, II, and III were treated. The protocol was different for each group of patients and included MT, the application of an ozone gel, an Er:YAG laser surgery session, conservative surgery with rotary/piezoelectric instruments, or surgical treatment, and then the monitoring of healing for at least 12 months. The protocols were performed once a week until complete recovery. Patients were reassessed weekly for the first month after treatment, monthly for the following quarter, and then every 3 months until the end of one year. The radiographic surveys were carried out 6 and 12 months after the last treatment. All of the patients in Group 4 (treated with medical therapy + ozone therapy + surgical treatment + laser Er:YAG) achieved complete clinical and radiographic recovery (100%) with complete remission of osteonecrosis. The proposed combined treatment of ozone therapy using laser Er:YAG and the MT + surgical treatment allowed us to obtain excellent results in the resolution of MRONJ. This success was explained by a series of characteristics specific to laser technology; in fact, thanks to its photoacoustic, photochemical, photothermal, and photomechanical properties, the laser made it possible to reduce the bacterial load at the intervention site.

**Keywords:** MRONJ; bisphosphonates; oral surgery; ozone; Er:YAG laser

## 1. Introduction

Medication-related osteonecrosis of the jaw (MRONJ) represents a pathological condition that affects the maxillary bones and it is disabling and highly degenerative. The three elements that allow the diagnosis of this condition [1] are an anti-angiogenetic or anti-absorption treatment that is in progress or occurred prior; exposed bone or probable bone by one or more intra- or extra-oral fistulas in the maxillofacial region for more than 8 weeks; and no radiotherapy in the head–neck region in the medical history [1].

In accordance with the 2014 American Association of Oral and Maxillofacial Surgeons (AAOMS) position paper, the main objectives of the treatment of patients at risk or with MRONJ are the following: priority of anticancer therapy with anti-resorption/anti-angiogenic drugs for the resolution of MRONJ, preservation of quality of life through patient education and reassurance, training in dedicated oral hygiene, being invited to frequent follow-up visits to prevent a recurrence, pain control with pharmacotherapy, and the prevention of lesion extensions and development of new osteonecrotic lesions [1].

At this time, medical or surgical therapies are proposed. Conservative or medical therapy consists of eliminating infectious foci through professional and domestic hygiene, the resolution of oral pathogenic foci, and the administration of local or systemic antibiotics. In addition, conservative therapies of recent clinical use are indicated.

For example, ozone therapy increases erythrocyte concentration and hemoglobin levels, stimulates diapedesis and phagocytosis, and has germicidal and analgesic effects. Ozone therapy can be combined with pre and intraoperative therapies [2].

Another conservative therapy is photobiomodulation (PBM), which has a biostimulating action, promotes the healing process in the peri-lesional tissues, and has antibacterial action [3]. PBM can be associated with pre- or intraoperative surgery [4]. Surgical therapy consists of removing the necrotic bone parts, which frequently show secondary infections.

Depending on the invasiveness, it is possible to distinguish between:

- Surgery using a conservative approach: surgical debridement with bone cutting, piezoelectric surgery, and laser-assisted surgery;
- Resective and reconstructive surgery: Allows a wide bone resection with healthy margins with reeducation by flaps with their own vascularization.

A systematic review and meta-analysis by Momesso et al. [5], whose aim was to evaluate the efficacy of laser therapy in MRONJ treatment, reported that qualitative data showed that treatment with laser surgery (Er:YAG) was the most effective regarding complete healing of the lesion (90%) compared with other treatments. Meta-analysis data showed that low-level laser therapy (LLLT) was more effective than medical treatment ($p = 0.006$), and surgical treatment was more effective than LLLT ($p = 0.008$). It appears that laser surgical therapy is a great management strategy for stage II MRONJ treatment.

The AAOMS in 2022 [6] stated that medical treatment is for stages I and II and surgical therapy is reserved for stage III or for those patients for whom medical therapy has not led to an effective improvement in symptoms (stage II) [1]. Holzinger D. [7] considered surgical protocols for stages I and II osteonecrotic lesions with excellent results. The surgical approach allowed for the ablation of necrotic tissues that do not allow wound healing [8].

The search for a minimally invasive surgical approach has increased in recent years. Among the minimally invasive surgical protocols, the use of laser surgery, as studied by Vescovi et al. [9] and by Stübinger et al. [10], is widely recognized in the international literature.

## 2. Materials and Methods

In this study, 52 patients were chosen among the patients of the dental oncology department treated between January 2019 and July 2020 at the Dental Clinic of Milan-Bicocca University. These patients were then retrospectively divided randomly into 5 groups based on a series of treatments known to cure MRONJ: 11 were treated with medical therapy only; 9 with medical therapy and conservative surgery with rotary instruments/piezosurgery; 14 with medical therapy and ozone therapy; 11 with medical therapy and Er:YAG laser therapy; 7 with medical therapy, ozone therapy, and Er:YAG laser (Table 1). The latter group is of the greatest interest to this study. The decision to opt for one therapy or another was arbitrary, depending on the operator.

**Table 1.** Group distribution of the patients, the number of patients treated in each group, and their therapies.

| Group | No. of Patients | Therapy |
|---|---|---|
| Group 1 | 11 | Medical therapy (MT) |
| Group 2 | 9 | MT + conservative surgery with rotary instruments/piezo-electric |
| Group 3 | 14 | MT + ozone therapy |
| Group 4 | 11 | MT + surgical treatment + laser Er:YAG |
| Group 5 | 7 | MT + ozone therapy + surgical treatment + laser Er:YAG. |

In reference to the 7 patients in Group 5 with MRONJ (range 44–85 years, mean 69 years, Table 2) stages I, II, and III, remote and current medical and dental anamnesis were collected for each patient, and each lesion was staged by clinical and radiographic

investigation according to the classification of Ruggiero and the American Association of Oral and Maxillofacial Surgeons (AAOMS) [1].

**Table 2.** Group 5 patients' demographic and clinical details.

| Patient | Age | MRONJ Stage |
| --- | --- | --- |
| Patient 1 | 44 | I |
| Patient 2 | 79 | II |
| Patient 3 | 61 | II |
| Patient 4 | 69 | I |
| Patient 5 | 85 | III |
| Patient 6 | 77 | I |
| Patient 7 | 65 | I |

Inclusion criteria:

- Presence of MRONJ

Exclusion criteria:

- Osteonecrosis of the jaw not exclusively linked to drugs (radiotherapy of the head–neck ORN region);
- Uncompensated systemic pathologies;
- Cancer pathology contraindicating the intervention.

The protocol provided for a specific organization chart; initially, MT was performed, followed by 8 applications of ozone and an Er:YAG laser surgery session, and finally, the monitoring of healing for at least 12 months.

MT included the dressing of the osteonecrotic lesions every 7 days in hospital, which consisted of irrigation with a physiological solution and the application of a 0.8% hydrogen peroxide ($H_2O_2$) gel for 15 min and homemade dressings, where each patient irrigated the wound with saline every 12 h at the end of each meal. Patients or family members were then asked to dress the wound, where a 0.8% hydrogen peroxide gel was applied for 15 min once a day after cleansing the bone exposure with a physiological solution. Antibiotics were administered intermittently or continuously in the cases of peri-lesional soft tissue superinfection with suppuration and pain symptoms (MRONJ stages II, III). Adequate and prolonged perioperative pharmacological treatment was of fundamental importance in order to minimize the risk of superinfection. The recommended antibiotic therapy was amoxicillin clavulanate in tablets of 1 g every 12 h + metronidazole in tablets of 250 mg every 8 h in order to acquire complete antibiotic coverage against anaerobic bacteria. The duration of administration was 7 days in the case of exclusive medical treatment and 14 in the case of combined treatment, with surgery on the seventh day.

For ozone therapy, after the dressing, an ozone gel of 15% (Ozoral gel, Innovares srl, Reggio Emilia, Italy) was applied to the wound for 10 min every 7 days. At the same time as the first treatment, an impression was taken for each patient to create a support for subsequent ozone gel applications. Not all patients consented to the support being made so sterile gauze covering the gel was used to prevent evaporation. Each session lasted 8 to 10 min under continuous suction to avoid gas inhalation.

Er:YAG laser surgery used the same medical preparation protocol, but the debridement in this case was treated with an Er:YAG laser set with an energy of 400 mJ; frequency of 13 Hz; VSP mode; fluence of 50 J/cm$^2$ above 300 mJ; irrigation with distilled water. The treatment sessions were repeated for up to one year of follow-up.

The protocol (Figures 1–6) was performed once a week until complete recovery.

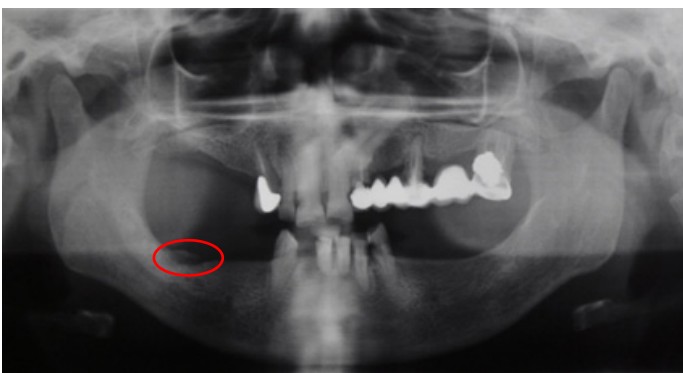

**Figure 1.** Radiographic image (orthopantomography) of the patient's initial situation; the osteonecrosis area is circled in red.

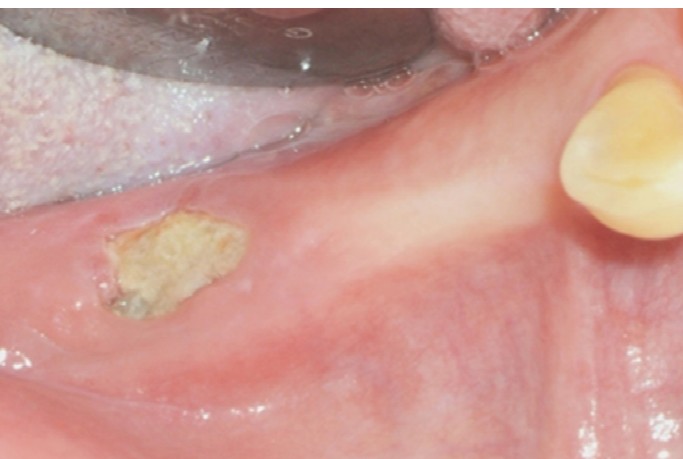

**Figure 2.** Intraoral image of the original situation of an osteonectotic lesion of the IV quadrant.

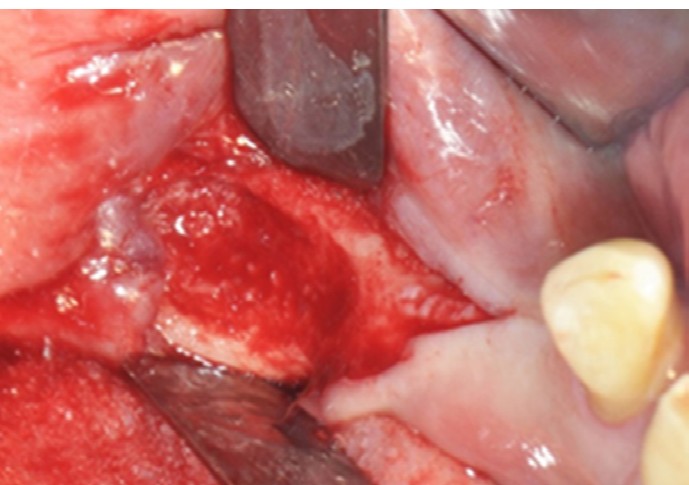

**Figure 3.** Intraoral intraoperative image of the elevation of the flap to full thickness.

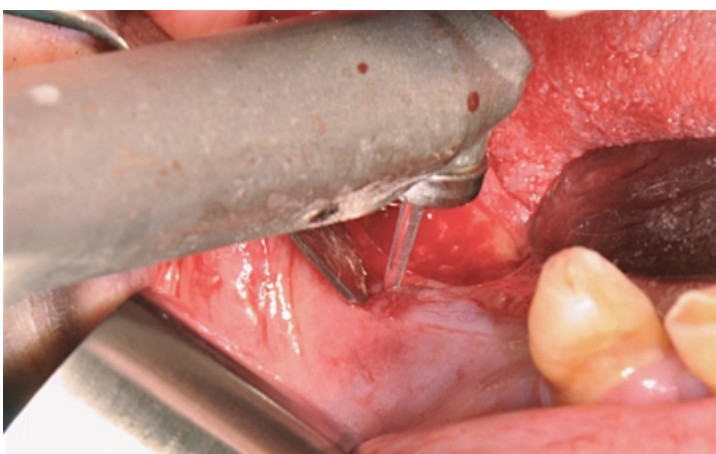

**Figure 4.** Intraoral intraoperative image of ablative surgery with ER:YAG laser.

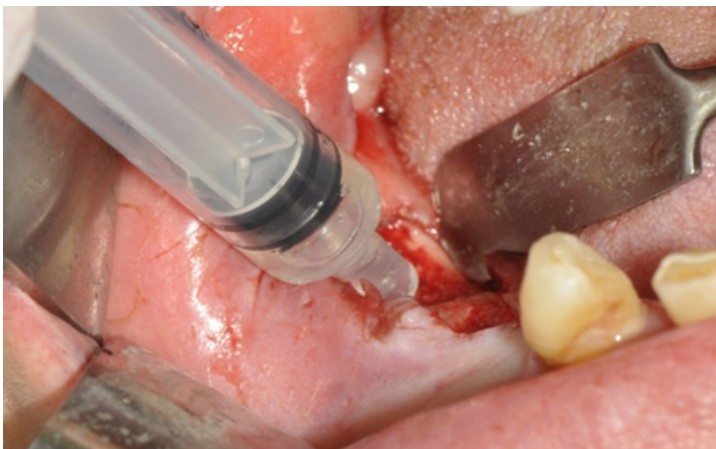

**Figure 5.** Intraoral intraoperative image of topical application of ozonated gel in the peri-operative phase.

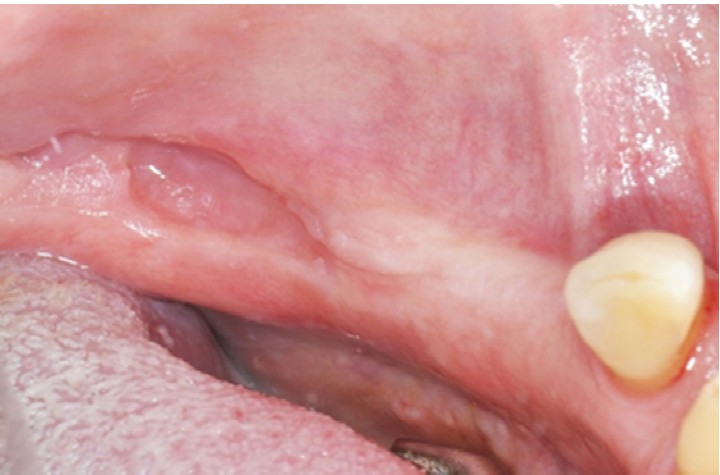

**Figure 6.** Complete healing of the lesion.

The results from the group of 7 patients treated with the protocol described above were compared with the results obtained from the other 4 groups with different therapies by colleagues in our department. Statistical analyses were performed with the Wilcoxon signed-rank test and the Chi-square test.

## 3. Results

All seven patients treated with MT + ozone therapy + Er:YAG laser surgery were reassessed weekly for the first month after treatment, monthly for the following quarter, and then every 3 months until the end of one year. Subsequently, half-yearly follow-ups were set up for the maintenance of oral hygiene. The radiographic surveys were carried out 6 and 12 months after the last treatment.

The healing was divided into:

- Complete healing of the wound, without any more bone exposure (if maintained for at least 12 months);
- Incomplete healing with or without stage reduction.

All treated patients achieved complete clinical and radiographic recovery (100%).

Tables 3–5 show the differences between the proposed treatment and those described in the Section 2.

**Table 3.** Complete and incomplete healings for each group.

| Group | No. | Incomplete Healings | % Incomplete Healings | Complete Healings | % Complete Healings |
|---|---|---|---|---|---|
| Group 1 | 11 | 5 | 45% | 0 | 0% |
| Group 2 | 13 | 3 | 15% | 5 | 38.5% |
| Group 3 | 9 | 1 | 11% | 7 | 78% |
| Group 4 | 11 | 2 | 18% | 9 | 82% |
| Group 5 | 7 | 0 | 0% | 7 | 100% |

**Table 4.** Wilcoxon test results applied to each group.

| Group | Wilcoxon Signed-Rank Test |
|---|---|
| Group 1 | significant treatment ($\leq$0.05) |
| Group 2 | significant treatment ($\leq$0.05) |
| Group 3 | significant treatment ($\leq$0.05) |
| Group 4 | significant treatment ($\leq$0.05) |
| Group 5 | significant treatment ($\leq$0.05) |

**Table 5.** Comparison of groups with the Chi-squared test. Values for which there were no statistically significant differences are highlighted in red.

| | Group 2 | Group 3 | Group 4 | Group 5 |
|---|---|---|---|---|
| Group 1 | 0.030613 | 0.00094 | <0.000001 | 0.000123 |
| Group 2 | - | 0.055086 | 0.019369 | 0.027605 |
| Group 3 | - | - | 0.5657 | 0.8415 |
| Group 4 | - | - | - | 0.8065 |
| Group 5 | - | - | - | - |

## 4. Discussion

Several protocols have been proposed in the literature because to date, there has been no common consensus. According to statements by the AAOMS in 2009, 2014, and 2022 [6], medical treatment is considered the first choice for stages I and II; surgical therapy, on the other hand, is reserved for stage III patients or for patients for whom medical therapy has not led to an effective improvement in symptoms (stage II) [1]. Despite this, surgical protocols have also been considered in the literature for stages I and II osteonecrotic lesions, with excellent results and higher percentages of partial and complete cures [7]. The surgical approach indeed allows for the ablation of necrotic tissues which, having no regenerative and restorative capacity, interfere with the healing of wounds [8]. A recent systematic review of the literature by Fliefel et al. [6] showed that the use of superficial surgical

protocols such as debridement is the most applied therapeutic choice. The search for a minimally invasive surgical approach allows for the better preservation of healthy bone tissue, avoiding excessive weakening of the jaw bones and allowing subsequent mobile prosthetic rehabilitation.

Among the minimally invasive surgical protocols, the use of laser surgery provides interesting results. As was shown in this study, patients treated with laser surgery had a 100% complete cure rate; these data confirm what was reported in the international literature [9,10]. This success is explained by a series of characteristics specific to laser technology; in fact, thanks to its photoacoustic, photochemical, photothermal, and photomechanical properties, the laser makes it possible to reduce the bacterial load at the intervention site. In addition, with laser surgery, a "smear layer" is not produced on the bone surface, which is a typical situation with rotary instruments; the vaporization of water promotes a cleaning effect on the surgical site that allows for the rapid removal of the treated tissue. The result is an extremely clean operating field. Another important feature is the biostimulatory property of the laser; at the bone level, it promotes the healing process in the post-surgical period [11]. There are several works in the literature about the efficacy of lasers and LEDs, not only in oral surgery but also in other fields of dentistry, especially in the management of periodontitis and peri-implantitis, and in orthodontics [12–21].

Data from the literature suggest excellent efficiency of the application of ozonated gel both when performed individually and in combination with surgical therapy. Ozone therapy was also performed in this study. The various properties of ozone therapy include an antimicrobial effect against anaerobic and aerobic bacteria, fungi, and viruses [22]; the stimulation of hemoglobin and red blood cell production with a relative increase in blood oxygenation; the regulation of cytokines involved in the immune response; increased phagocytosis and diapedesis; and the stimulation of angiogenesis and fibroblasts. Clinically, ozone therapy induces the formation of a bone seizure, stimulates the vascularity of the underlying bone, and stimulates the formation of granulation tissue. Therefore, the removal of these lesions exposes the underlying bone tissue in the regeneration phase to the oral environment, which is not necessary for surgical treatment [23]. Agrillo et al. [24] used topical ozone therapy in the treatment of MRONJ, achieving the remission of symptoms in 90% of cases, and Ripamonti et al. [25] also applied this combined method to medical therapy for the only conservative treatment of osteonecrotic lesions of less than 2.5 cm, achieving resolution with complete healing of all lesions analyzed. The results obtained in this study confirm that, if combined with laser surgery, 100% complete cures can be achieved. Therefore, ozone therapy can be combined with medical therapy in the initial management of MRONJ and can be used in the preparation of surgical treatment in the intraoperative phase, after having performed the complete ablation of the lesion, and in the post-surgical healing phase. Thanks to the properties listed above, ozone therapy can also be used in patients with stages II and III MRONJ, where, due to oncological problems, a surgical approach is not allowed.

It is important to consider other treatments; in fact, Paulo et al. [26] reproduced in vivo a model of MRONJ in Wistar rats. After tooth extraction, biphasic calcium phosphate (BCP) granules were placed in the alveolus. After every evaluation (through nuclear medicine, radiology, macroscopic observation, and histologic analysis), the authors observed that calcium phosphate ceramics were able to limit zoledronate toxicity in vivo and favor healing, which was evidenced by medical imaging (nuclear medicine and radiology), macroscopically, and through histology. The studied therapeutic option presented itself as a potential solution to prevent the development of maxillary osteonecrosis.

Ye, P. et al. [27] conducted a meta-analysis involving a systematic search of PubMed, EMBASE, Wiley Online Library, and the Cochrane Library for eligible studies from their inception to November 2019 in accordance with preselected criteria. Thirteen studies that investigated APCs in the treatment of MRONJ were eligible for inclusion in the meta-analysis of 223 patients and 33 lesions. The results suggested that the use of APCs is a promising therapeutic regimen, as it provides additional benefits to surgery in the

treatment of MRONJ. To achieve the benefits, a tension-free primary closure of the soft tissue is also recommended. Randomized studies with large sample sizes are warranted to confirm our findings.

This study has many limitations, as we did not have the opportunity to assess other treatments such as autologous platelet concentrates, calcium phosphate ceramics such as hyperbaric oxygen, auto/tetracycline fluorescence-guided bone surgery, medical drugs such as teriparatide, or a combination of pentoxifylline and tocopherol, which are all important and should be assessed further to see which method is the best option for a patient.

## 5. Conclusions

Considering what has been highlighted by this study, it can be said that conservative surgical treatment is also a valid choice in stages I and II lesions and that laser surgery combined with ozonated gel is extremely effective in the treatment of MRONJ.

These statements are consistent with the international literature, where numerous articles affirm the success of a surgically conservative approach in association with supportive therapy (laser therapy or ozone) to improve surgical management.

**Author Contributions:** Conceptualization, G.P.; Data curation, P.C.; Formal analysis, G.P.; Investigation, G.P.; Methodology, G.C.; Project administration, G.C.; Resources, G.C.; Software, A.A.B.; Supervision, P.C.; Validation, G.C.; Visualization, A.A.B.; Writing—original draft, G.P.; Writing—review and editing, P.C. and A.A.B. All authors have read and agreed to the published version of the manuscript.

**Funding:** This research received no external funding.

**Informed Consent Statement:** Informed consent was obtained from all subjects involved in the study.

**Data Availability Statement:** Not applicable.

**Conflicts of Interest:** The authors declare no conflict of interest.

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
