# Peer review of "Treatment of Medication-Related Osteonecrosis of the Jaw (MRONJ) with Er:YaG Laser and Ozone Therapy: A Case Series"

_inventions, doi:10.3390/inventions7040097_

Round 1

Reviewer 1 Report

ABSTRACT

“The purpose of this study is to evaluate the efficacy of the combination of ozone gel and Er:YAG laser in the treatment of Medication-Related Osteonecrosis of Jaws”.  Abstract is too summary and too confusing.

Line 10-11: “ …7 patients 10 with MRONJ of stages I, II and III were treated…”

This sentence is out of context.

In the materials and methods (line 52) the authors says:”…In this study, 52 patients treated between January 2019…”.

Were 7 or 52 patients treated?

INTRODUCTION

MATERIALS AND METODHS

Line 66-69: Os autores dividiram os doentes em 5 grupos. “...11 were treated with medical therapy only, 9 with medical therapy and conservative surgery with rotary instruments / piezosurgery, 14 with medical therapy and ozone therapy, 11 with medical therapy and Er: YAG laser therapy, 7 with medical therapy, ozone therapy and Er: YAG laser….”

How was the selection and distribution of patients carried out among the different groups?

Line 98: “…Adeguate and prolonged perioperative…”

Do you mean adequate?

Was sutured after surgery?

RESULTS

Line 172-77: “…The latter 7 patients with drug-related osteonecrotic maxillary lesions (MRONJ) (range 44-85 years, mean 69 years) of stages I, II and III were selected according to the following criteria. Near and distant medical and dental histories were collected for each patient. Each lesion was staged by clinical and radiographic investigation in accordance with the classification of Ruggiero and American Association of Oral and Maxillofacial 76 Surgeons (AAOMS). …”

The relevance of this sentence is not understood.

Table 2.

This table needs more information in the title.

Figure 1. Radiographic image of the starting situation.

The legend needs more information about the type of lesion and the radiographic location.

There are no photos of post-operative controls.

DISCUSSION

Other treatments such as autologous platelet concentrates, hyperbaric oxygen, Auto/tetracycline fluorescence-guided bone surgery, Calcium Phosphate Ceramics (Materials. 2020 Apr 22;13(8):1955.  doi: 10.3390/ma13081955.), medical drugs like teriparatide or the combination between pentoxifylline and tocopherol, or less well documented and known considering their clinical effectiveness should be discussed.

lines 190-193, the actions of ozone should be linked with the physiopathology of MRONJ.

An example of how this study has many limitations .

REFERENCES

Reference 6 should be revised and 19 and 20 have nothing to do with the topic. Recent bibliography must be included.

Author Response

Dear reviewer,

In responses to your kind review, thank you for your work on our article ' Treatment of Medication-Related Osteonecrosis of Jaws (MRONJ) with Er:YaG Laser and Ozone Therapy: A Case Series'.

We changed the file according to your indications and as follows:

  • Abstracts:
    • We revised the abstract trying to follow your indications
    • There were 52 patients treated, and only 7 of them was in the group 4 and were treated with the protocol we are presenting in this paper. We changed that part, we tried to make it comprehensible
  • Materials and methods:
    • How was the selection and distribution of patients carried out among the different groups? The selection was random.
    • We have reviewed and corrected the errors
  • Results:
    • We have reviewed and corrected the errors
    • Table 2: We added more information in the title.
    • In the legend we added more information about the type of lesion and the radiographic location.
  • Discussion
    • We reviewed that part, and included 2 articles about APC and BPC (the one you kindly suggested).
    • We added a part about the limitations of this study.
  • References
    • We checked it, and reviewed

We hope that our paper will meets your expectations and will be published on Inventions journal.

Thank you for you time.

Dr. P. Caccianiga

Dr. A.A. Bader

Reviewer 2 Report

1.      The abstract is not structured! Clinical case descriptions demand it!

2.      Only 5 authors are mentioned in the introduction, the rest of them (to 25) are directly in the  Discusion part. To approve the quality of the article all of them should be cited in the introduction, too.

3.      The materials and methods are very well visualized in tables and figures.

4.      The results are clearly shown in tables.

5.      The discussion is voluminous enough.

6.      The conclusion is clear and shows the main conclusions of the study.

7.      References are sufficient number for case report.

Author Response

Dear reviewer,

In responses to your kind review, thank you for your work on our article 'Treatment of Medication-Related Osteonecrosis of Jaws (MRONJ) with Er:YaG Laser and Ozone Therapy: A Case Series'.

We changed the file according to your indications and as follows:

  • We modified the abstract, hope now it’s better.
  • We reviewed the introduction.

We hope that our paper will meets your expectations and will be published on Inventions journal.

Thank you for you time.

Dr. P. Caccianiga

Dr. A.A. Bader

Round 2

Reviewer 1 Report

The authors substantially corrected the article, so I understand that it can be published